# Impairment of Nucleolin Activity and Phosphorylation by a Trachylobane Diterpene from *Psiadia punctulata* in Cancer Cells

**DOI:** 10.3390/ijms231911390

**Published:** 2022-09-27

**Authors:** Maria Laura Bellone, Lorenzo Fiengo, Carmen Cerchia, Roberta Cotugno, Ammar Bader, Antonio Lavecchia, Nunziatina De Tommasi, Fabrizio Dal Piaz

**Affiliations:** 1Department of Pharmacy, University of Salerno, Via Giovanni Paolo II, 84084 Fisciano, Italy; 2“Drug Discovery” Laboratory, Department of Pharmacy, University of Napoli “Federico II”, Via D. Montesano, 49, 80131 Napoli, Italy; 3Department of Pharmacognosy, Faculty of Pharmacy, Umm Al-Qura University, Mecca 21995, Saudi Arabia; 4Department of Medicine, Surgery and Dentistry, University of Salerno, Via S. Allende, 84081 Baronissi, Italy

**Keywords:** plant diterpenes, nucleolin, protein–drug interaction, cancer cell apoptosis, post-translational modification

## Abstract

Human nucleolin (hNcl) is a multifunctional protein involved in the progression of various cancers and plays a key role in other pathologies. Therefore, there is still unsatisfied demand for hNcl modulators. Recently, we demonstrated that the plant ent-kaurane diterpene oridonin inhibits hNcl but, unfortunately, this compound is quite toxic for healthy cells. Trachylobane diterpene 6,19-dihydroxy-ent-trachiloban-17-oic acid (compound **12**) extracted from *Psiadia punctulata* (DC.) Vatke (Asteraceae) emerged as a ligand of hNcl from a cellular thermal shift assay (CETSA)-based screening of a small library of diterpenes. Effective interaction between this compound and the protein was demonstrated to occur both in vitro and inside two different types of cancer cells. Based on the experimental and computational data, a model of the hNcl/compound **12** complex was built. Because of this binding, hNcl mRNA chaperone activity was significantly reduced, and the level of phosphorylation of the protein was affected. At the biological level, cancer cell incubation with compound **12** produced a cell cycle block in the subG0/G1 phase and induced early apoptosis, whereas no cytotoxicity towards healthy cells was observed. Overall, these results suggested that 6,19-dihydroxy-ent-trachiloban-17-oic could represent a selective antitumoral agent and a promising lead for designing innovative hNcl inhibitors also usable for therapeutic purposes.

## 1. Introduction

Human nucleolin (hNcl) is a 709 amino acid protein that can represent as much as 10% of total nucleolar proteins [1]. Structurally, it is organized into three domains: the N-terminal composed of highly acidic regions interspaced with basic sequences and showing several phosphorylation sites, the central region consisting of four RNA-binding domains (RBDs) and the C-terminal domain, which contains high levels of NG, NG-dimethylarginines [2]. hNcl establishes different interaction networks that confer several functions to it. In the nucleolus, it is directly involved in ribosome biogenesis [3], chromatin remodeling [4], transcription regulation [5], and telomerase activity [6]. In the nucleoplasm, hNcl interacts with several RNAs and proteins (e.g., transcription factors) and is involved in the regulation of the cellular response to stress [7]. Furthermore, it constantly shuttles between the nucleus and cytoplasm, allowing for the import/export of ribosomal proteins and post-transcriptional and translational regulation of various mRNAs, including p53, B-cell lymphoma 2 (Bcl-2), and serine/threonine kinase 1 (Akt) [8]. For this latter function, hNcl is considered a sort of RNA molecular chaperone. Finally, cell-surface hNcl behaves as a receptor, binding to diverse proteins, thus inducing tumorigenesis (cell migration, adhesion, angiogenesis) and mediating inflammation and viral infections [9]. This direct correlation between function and cellular localization implies that hNcl trafficking, strictly dependent on its phosphorylation, is significantly altered by some pathological conditions [10]. Moreover, hNcl makes a significant contribution to cancer progression through different mechanisms, possibly the most prominent being the protection of the Bcl-2 and Akt mRNAs from hydrolysis, resulting in an increased intracellular abundance of the two anti-apoptotic proteins, the stabilization of the micro RNAs affecting the expression of tumor suppressor genes [11] and the interaction at the plasma membrane with proteins promoting cell survival, proliferation, and migration or angiogenesis [12,13,14,15,16,17,18,19]. Therefore, an increasing interest in hNcl inhibitors has been observed, leading to the identification of a few compounds interacting with membrane-located hNcl [10] the most efficient being the AS1411 aptamer [20] and the F3 peptide [21] and the pseudopeptide N6L [22]. However, there is a largely unmet demand for compounds capable of modulating the activity of this protein in other cellular compartments [23].

We recently identified the plant ent-kaurane diterpene oridonin as an inhibitor hNcl [24]. This compound, well known for its anti-cancer but also toxic effects [25,26], was able to interact with the protein in human cervical cancer (HeLa) and human T-lymphocyte (Jurkat) cells, impairing the mRNA chaperone and the ribosome biogenetic activity of hNcl [24]. Based on these results, we attempted to identify compounds structurally related to oridonin capable of interacting with hNcl and modulating its activity in cancer cells but possibly showing negligible toxicity towards health cells. We screened a small library of diterpenes using a cellular thermal shift assay (CETSA) [27] and the obtained results led to the identification of the plant trachylobane diterpene 6,19-dihydroxy-ent-trachiloban-17-oic acid (compound **12**) as a putative hNcl ligand. The interaction of hNcl with this ligand was thus investigated to define the structure of the resulting complex and to evaluate its effects on cancer cells.

## 2. Results and Discussion

### 2.1. CETSA-Based Screening of Putative hNcl Interactors

To identify new potential hNcl modulators, a small library of diterpenes was subjected to CETSA-based screening to assess their affinity to hNcl in the cell. As we have previously described oridonin as an effective hNcl inhibitor, compounds that share structural characteristics with this plant diterpene were selected. Specifically, ten ent-kaurane and six ent-trachylobane diterpenes were chosen (Appendix A). Oridonin was used as a positive control.

Based on previously published results, CETSA experiments were carried out using a Jurkat cell line using the in-cell hNcl denaturation temperature of 55 °C. The efficacy of the different testing compounds in preventing or reducing hNcl aggregation at that temperature was monitored by Western blot (Appendix A). Four of the sixteen tested compounds protected hNcl through CETSA (Figure 1A), but only a trachylobane diterpene (6,19-dihydroxy-ent-trachiloban-17-oic acid) extracted from *Psiadia punctulata* (Vake) Asteraceae (**12**) (Figure 1B) produced an effect higher than that of the positive control.

### 2.2. Validation of the Interaction of Compound 12 with hNcl

The MTT-based assay demonstrated that compound **12** inhibited the proliferation of two different cancer cell lines, showing IC_50_ values of 50 μM and 25 μM against Jurkat and HeLa cells, respectively. No toxicity of this compound against PBMC and HaCat cell lines was observed up to a concentration of 100 μM. Therefore, subsequent experiments were conducted on the HeLa cell line. To confirm that hNcl was a target of compound **12** in the cancer cells, DARTS experiments [28] were carried out. The obtained results (Figure 1C) showed that incubation with compound **12** induced a significant increase in hNcl resistance to protease action. 

The **12**/hNcl interaction was also investigated using CETSA and its effect on the hNcl denaturation temperature (Tm) in the cells was analyzed. HeLa cells incubated with **12**, oridonin or the vehicle were heated to different temperatures in the 38–61 °C range, and the amount of soluble hNcl remaining at each T value was measured. The resulting hNcl vs. T graphs (Appendix A and Figure 1D) indicated that **12** stabilized hNcl more than oridonin did, and the Tm observed for the protein in the cells incubated with **12** was estimated at 55.5 °C (Appendix A and Figure 1D). Further CETSAs were performed using various concentrations of the compound (1–20 μM) and the amount of hNcl resisting thermal denaturation at 55.5 °C was then reported in a graph as a function of the concentration of **12** used (Appendix A and Figure 1E). This experiment allowed the calculation of an EC_50_ (the compound concentration producing half of the maximum effect on hNcl stabilization) of 5 μM. Finally, using surface plasmon resonance (SPR), a dissociation constant (K_D_) of 58 ± 3 nM was measured for the in vitro interaction of compound **12** with immobilized hNcl. The shape of the sensorgram (Appendix A) and the measured k_off_ (0.0037 ± 0.005 s^−1^) suggested the formation of a highly stable complex with 1:1 stoichiometry.

Moreover, to obtain information concerning the protein region involved in the interaction with **12**, a new SPR analysis was carried out using an hNcl fragment consisting of RBD1 and 2 (RBD1,2). Elaboration of the resulting sensorgrams (Appendix A) demonstrated that the interaction between **12** and RBD1,2 resembled that with the whole protein in terms of both K_D_ (74 ± 5 nM) and k_off_ (0.0053 ± 0.009 s^−1^).

### 2.3. Molecular Modeling of the 12/hNcl Complex

We undertook docking studies using Glide to describe the interaction between **12** and RBD1,2. RBD1,2 form a molecular clamp to bind the RNAs and are reported targets of the aptamer AS1411 [29,30,31,32] and small-molecule natural products [33]. Both RBD1,2 adopt a βαββαβ fold, common to other RBDs, and are connected by a flexible linker that becomes more structured upon RNA binding [34]. We initially identified druggable cavities on RBD1,2 that might be able to bind to **12** using the site-detecting software SiteMap [35,36] on the ensemble of 20 NMR structures of RBD1,2 from human nucleolin (Protein Data Bank ID: 2KRR) [29]. A SiteScore cut-off of 0.8 was employed to focus the docking calculations only on the most relevant sites [35,36] using an ensemble-based approach and the best-scored pose of **12** was selected to represent the ligand-binding mode [37]. As shown in Figure 2A, compound **12** is hosted in a 288 Å^3^ cavity lodged between the linker and RBD2. The carboxylic moiety of **12** established an H-bond and a tight salt bridge with K81 (Figure 2B,C). Moreover, the OH group at position 6 forms an H-bond with the CO backbone of P82 and, in turn, accepts an H-bond from the NH backbone of G84. In addition, the OH group at position 19 engages further H-bonds with the side chain of D92 and the NH backbone of G166. The ent-trachiloban skeleton forms additional hydrophobic interactions with the side chains of P168 and K81. Our results underscore that the **12**/RBD1,2 interaction is critical, thus suggesting that compound **12** might impair the stabilization and further processing of RNA.

To gain insights into the possible binding mode of oridonin into RBD1,2 of nucleolin, we undertook docking studies using the same protocol adopted for **12**. In fact, both compounds have been found to impair the mRNA chaperone activity of nucleolin. Structurally, oridonin is an ent-kaurane-type diterpenoid bearing an α,β -unsaturated ketone function. This moiety has been reported to be crucial for the anticancer activity of oridonin and has the potential to react covalently with nucleophilic residues. However, in our case, SPR and MS-based peptide mapping previously performed ruled out this possibility, suggesting that oridonin binds to nucleolin through non-covalent interactions [24]. The ent-kaurane skeleton of oridonin fits well within the druggable site identified by SiteMap, engaging mostly hydrophobic interactions with the side chains of P168 and K81 (Appendix A). Oridonin hydroxyl group on ring A donates an H-bond to the backbone CO of P82, in a similar fashion to **12**; in addition, the carbonyl group of the exo-methylen-cyclopentanone moiety (D-ring accepts an H-bond from the backbone NH of K169 [24]. Overall, we observed fewer interactions formed by oridonin within the proposed nucleolin binding site in comparison with **12**, further confirming that **12** most effectively binds to hNcl, as also observed in CETSA assays.

### 2.4. Modulation of hNcl Activity by Compound **12**

A significant validation of the proposed model was sought through specific in-cell biological assays. hNcl is a multifunctional protein involved in a number of cell pathways, depending on its cellular localization. However, one of the most relevant roles probably played by this protein is mRNA chaperone activity carried out through the interaction of the protein RBDs with the 3′-untranslated region of several target mRNAs. As hNcl has been reported to bind the mRNAs encoding for Bcl-2 [11] and Akt [38] its inhibition can result in the destabilization of these two messengers. Accordingly, through real-time quantitative PCR, we found that in the HeLa cells treated with 20 μM compound **12** for 4 h and 8 h, the mRNA levels of Bcl-2 were time-dependently reduced (Figure 3A).

In addition, the number of messengers for Akt was lowered following cell incubation with compound **12**, although the effect was less evident for this mRNA. The consequences of the decrease in Bcl-2 and Akt mRNAs amounts on the levels of the two proteins were also assayed (Figure 3B), thus confirming that the treatment of HeLa cells with **12** significantly lowers the levels of Akt. 

The hNcls’ pivotal effect on ribosome assembly through the binding of the C-terminal glycine- and arginine-rich domains with ribosomal proteins and large rRNAs has been reported [39] and some hNcl inhibitors have been suggested to prevent this binding, thus significantly impairing protein expression [40]. Therefore, we compared the protein neo-syntheses in untreated HeLa cells and in HeLa cells subjected to incubation with 20 μM compound **12**, but no significant differences emerged (Figure 3B), suggesting the negligible effect of that compound on the interaction of hNcl with ribosomes. 

### 2.5. Biological Effects of Compound **12** Treatment on Cancer Cells

Finally, we determined whether the treatment with **12** could affect hNcl expression, phosphorylation, and cellular localization. Western blot analyses demonstrated that incubation of cancer cells with subtoxic amounts of the compound (20 μM) for 4, 6, or 16 h did not modify the hNcl levels, but the phosphorylation of the protein was inhibited after a 4 h treatment (Figure 4A,B).

Based on our data, it is not possible to assess whether this reduced phosphorylation depends on conformational changes that occur following the interaction of **12** with hNcl and involving the N-terminal domain of the protein, where the phosphorylation sites are located. However, it is interesting to note that a similar effect was also observed for AS1411, one of the most studied inhibitors of hNcl. Indeed, Iturriaga-Goyon et al. showed that cellular treatment with this aptamer significantly and selectively reduced the phosphorylation of hNcl, slightly influencing the total protein level [41].

The distribution of hNcl in the different cell compartments (nucleus, cytosol, and membrane) did not show dramatic variations following cell incubation with compound **12**. The observed decrement of phosphorylated hNcl (p-hNcl) seems to be attributable to the overall decrease in its levels rather than to a modification of its cellular distribution. 

Cytofluorimetry was used to investigate the effects of compound **12** on cancer cell cycle and apoptosis, and it showed that this compound allowed cell accumulation in the subG0/G1 phase in a dose-dependent manner (Figure 5A). Annexin V-FITC/PI was then used to study the effect of compound **12** on cell death; it was found that a 48 h treatment produced an approximately 20% increase in early apoptosis (Figure 5B), as confirmed by Parp and pro-Caspase 3 cleavage (Figure 5C). 

These results could be related to the effects that compound **12** exerts on nucleolin RNA chaperone activity and phosphorylation. It has indeed been shown that drug-induced downregulation of nucleolin causes cell apoptosis [42]. Furthermore, the ability of hNcl to protect tumor cells from apoptosis was strongly related to its phosphorylation, and it was shown that a reduction in the p-hNcl/hNcl ratio corresponded to a greater onset of the apoptotic process [43]. 

## 3. Materials and Methods

### 3.1. Materials

FBS was purchased from GIBCO (Life Technologies, Grand Island, NY, USA); we obtained from Abcam (Cambridge, UK) anti-nucleolin (rabbit polyclonal) and anti-phosphoT76-nucleolin (rabbit monoclonal) antibodies. We purchased from Santa Cruz Biotechnology (Santa Cruz Biotechnology, Inc., Delaware, CA, USA) the following antibodies: anti-Akt (rabbit polyclonal), anti-Bcl2 (mouse monoclonal), anti-Tubulin (mouse monoclonal), anti-Gapdh (glyceraldehyde-3-phosphate dehydrogenase) (rabbit polyclonal), anti-Caspase-3 (mouse monoclonal), anti-Hsp70 (mouse monoclonal) and anti-Caveolin (rabbit polyclonal), whereas anti-Lamin A (rabbit polyclonal) was purchased from Merk (USA). Anti-PARP (rabbit polyclonal) was purchased from Cell-Signaling (USA). From Jackson Immuno Research (Baltimore, PA, USA) we obtained appropriate peroxidase-conjugated secondary antibodies. Compounds **1–16** and oridonin were obtained from plant material according to the literature [24,44].

### 3.2. Cell Culture and Treatment

HeLa (cervical carcinoma) and Jurkat (human T lymphocyte) cell lines were purchased from the American Type Cell Culture (Rockville, MD, USA). The HeLa cell line was maintained in Dulbecco’s Modified Eagle Medium (DMEM) and Jurkat cell line in Roswell Park Memorial Institute Medium (RPMI 1640). The cells were cultured using 10% FBS, 100 mg/L streptomycin, and penicillin 100 IU/mL supplemented media, at 37 °C under a humidified atmosphere of 5% CO_2_. A logarithmic growth was obtained by subculturing the cells every 2 days. To set up the stock solutions, all the compounds were dissolved in DMSO and stored in the dark at 4 °C. Appropriate dilutions were prepared in culture medium immediately prior to use. In all experiments, the final concentration of DMSO did not exceed 0.1% DMSO (*v*/*v*). 

### 3.3. MTT Assay

HeLa cells were seeded at 1 × 10^4^ cells/well in 96-well plates. The day after the cells were incubated for 48 h with different concentrations of compounds (10–100) μM. The number of viable cells was quantified by MTT ([3-(4,5-dimethylthiazol-2-yl)-2,5-diphenyl tetrazolium bromide]) assay. Absorption at 550 nm for each well was assessed using a microplate reader (LabSystems, Vienna, VA, USA). Half maximal inhibitory concentration (IC_50_) values were calculated from cell viability dose–response curves and defined as the concentration resulting in 50% inhibition of cell survival as compared to controls.

### 3.4. CETSA-Based Screening

Jurkat cells were seeded at 4.0 × 10^5^ cells/mL. After 24 h, the cells were treated with diterpenes or the positive control oridonin (final concentration 20 μM) and then incubated for 2 h in a CO_2_ incubator at 37 °C. Afterward, the cells were collected and centrifuged, and the cell pellets were washed with PBS and gently suspended in 1 mL PBS supplemented with protease inhibitors. Each cell suspension (DMSO, positive control, and treated samples) was heated in a PCR machine (Invitrogen Life Science Technologies, Waltham, MA, USA) at 55 °C for 3 min. The cells were then lysed via freezing/thawing and centrifuged at 4 °C for 20 min at 20,000× *g*. Subsequently, the proteins were fractionated via SDS-PAGE and analyzed by Western blot. Densitometric analysis of the bands was performed using the ImageJ software. The results were plotted on graphs reporting the ratio between the density measured for each nucleolin band and the density measured for the corresponding Gapdh band. For the histogram plot, the expression level of the nucleolin protein was normalized to the negative control.

### 3.5. Determination of the Apparent Melting Temperature (Tm) of hNcl in HeLa Cells 

HeLa cells were seeded at 1.5 × 10^6^ in 100 mm culture dishes. After 24 h, the cells were treated with compound **12** or oridonin (final concentration 20 μM) and incubated for 2 h in a CO_2_ incubator at 37 °C. The cells were then collected and centrifuged, and the cell pellets were washed with PBS and gently suspended in 1 mL PBS supplemented with protease inhibitors. Each cell suspension (DMSO, positive control and treated samples) was divided into eight fractions, all of which were heated in a PCR machine (Invitrogen Life Science Technologies) at different temperatures (38.4, 41.3, 45.5, 52.8, 55.6, 57.6, 59.8 and 61.3 °C) for 3 min. The cells were then lysed and analyzed, as reported above.

### 3.6. Determination of the EC_50_


HeLa cells were seeded at 1.0 × 10^5^ in 12-well plates. The treatment was performed for 2 h using compound **12** at the following concentrations: 0.1, 0.5, 1, 2, 5, 10, 15, and 20 μM. Subsequently, the drug-containing media were removed through centrifugation; the cells were washed with PBS, aliquoted in 0.2 mL tubes, and then heated at 55.5 °C (Tm of hNcl) for 3 min. They were then lysed and analyzed, as reported above. The hNcl/Gapdh density ratio measured at the nucleolin concentration producing the maximum stabilizing effect was set as 100%.

### 3.7. Drug Affinity Responsive Target Stability (DARTS) Experiments

HeLa cells were seeded at 1.5 × 10^6^ in 100 mm culture dishes and treated with compound **12** (5 or 20 μM) or 0.1% DMSO for 2 h at 37 °C. The HeLa cells were lysed in RIPA buffer (10 mM Tris HCl pH 7.6, 1 mM EDTA, 1% Triton X-100, 0.1% sodium deoxycholate, 0.1% SDS, 140 mM sodium chloride and 1 mM PMSF) supplemented with phosphatase inhibitors. The samples underwent proteolysis with subtilisin (enzyme:lysate 1:6000 *w/w*) for 30 min, and hydrolysis was stopped by adding Laemmli buffer 4× and incubating the mixture at 95 °C for 5 min. Finally, the samples were fractionated via SDS-PAGE and analyzed by Western blot. Densitometry of bands was performed with the ImageJ software. Experiments were performed in triplicate.

### 3.8. Surface Plasmon Resonance

A Biacore 3000 optical biosensor was used to conduct Surface Plasmon Resonance (SPR) experiments. Proteins (hNcl and RBD 1,2) were dissolved at 100 µg/mL in 50 mM sodium acetate at a pH 4.5 and injected at a flow rate of 5 µL/min on a research-grade CM5 sensor chip (GE Healthcare, Chicago, IL, USA) to achieve immobilization. A standard amine-coupling protocol was used. Compound **12** was dissolved in 100% DMSO to obtain 4 mM solutions and diluted in 1:200 (*v*/*v*) in PBS (10 mM NaH_2_PO_4_, 137 mM NaCl, 2.7 mM KCl pH 7.4). The final concentration of DMSO was 0.1% (*v*/*v*). This stock solution was used as a starting point to produce a four-concentration points series (0.025, 0.2, 1, and 4 µM). Measurements were performed at 25 °C, using a flow rate of 5 µL/min flow rate and adopting an association time of 60 s; dissociation of the complex was monitored for 300 s. The resulting sensorgrams were elaborated through the BIAevaluation software (GE Healthcare). Thermodynamic constants were calculated by adequately fitting the experimental curves with a single-site bimolecular interaction model.

### 3.9. Docking Studies 

The core structure of compound **12** was retrieved from the Cambridge Structural Database (refcode: ADOFIJ) and modified with the fragment dictionary of Maestro. The ligand was then preprocessed with LigPrep and optimized by Macromodel, using the MMFFs force field with the steepest descent (1000 steps) followed by truncated Newton conjugate gradient (500 steps) methods. Partial atomic charges were computed using the OPLS-AA force field. The solution NMR structure of human nucleolin RBD1,2 (PDB ID: 2KRR) [29] was retrieved from the Protein Data Bank. The ensemble of 20 NMR structures was prepared using the Protein Preparation Wizard implemented in Maestro. Hydrogen atoms were added to the protein consistent with the neutral physiologic pH. Successively, the protein hydrogens only were minimized using the Impref module with the OPLS_2005 force field.

The site recognition software SiteMap was run on the 20 prepared structures of RBD1,2 to identify potential ligand-binding sites. Sites were kept if they comprised at least 15 site points. The standard grid (0.7 Å) for site points was used, together with the more restrictive hydrophobicity definition. For each of the identified sites, SiteMap calculates a SiteScore to assess a site’s propensity for ligand binding, accurately ranking possible binding sites to eliminate those not likely to be pharmaceutically relevant. Docking simulations of compound **12** and oridonin were then performed to find a binding pose on these pockets, following an ensemble docking approach.

A docking grid was generated with Glide enclosing a box centered on the residues bordering the most promising sites identified by SiteMap, with an inner box size of 10 × 10 × 10 Å and an outer box of 30 × 30 × 30 Å. A scaling factor of 0.8 was set for van der Waals radii of receptor atoms. Ligand sampling was allowed to be flexible. Default docking parameters were used, and no constraints were included. The results of the calculations were evaluated and ranked based on the Glide SP scoring function [37,45]. The final receptor–ligand complex was chosen by selecting the highest scoring pose within the sites identified by SiteMap. Figures were generated using Pymol (Version 2.0).

### 3.10. Cytosol, Membrane, and Nuclear Extracts

For cytosol and membrane protein extraction, the HeLa cells were seeded at 1.5 × 10^6^ in 100 mm culture dishes and incubated with oridonin (5 μM) or compound **12** (20 μM) for 4 h at 37 °C. The cells were harvested in PBS and centrifuged at 4 °C for 5 min at 600× *g*. Subsequently, the cell pellets were resuspended in lysis buffer (20 mM Tris HCl pH 7.4, 250 mM sucrose, 1 mM dithiothreitol (DTT), protease inhibitors, and 1 mM EDTA in water), sonicated (5 s pulse–9 s pause for 2 min, amplitude 42%) and then centrifuged at 4 °C for 10 min at 5000× *g*. The obtained supernatants were ultra-centrifuged at 4 °C for 1 h at 100,000× *g*, and cytosolic extracts were obtained. Each resulting pellet was resuspended in a lysis buffer and ultracentrifuged at 4 °C for 1 h at 100,000× *g*. The pellets were then resuspended in 250 μL solubilization buffer (20 mM Tris HCl pH 7.4, 1 mM DTT, 1 mM EDTA, and 1% Triton X-100 in water) and left overnight on an orbital shaker at 4 °C. After that, the solution was centrifuged at 4 °C for 30 min at 50,000× *g*, and membrane proteins were obtained.

To achieve nuclear fraction, the pellets—obtained as described above—were resuspended in 500 μL buffer A (10 mM Hepes pH 7.9, 1 mM EDTA pH 8.0, 60 mM KCl, 0.2% NP-40, 1 mM DTT, 1 mM PMSF and protease inhibitors) and incubated on ice for 10 min. The samples were then centrifuged at 4 °C for 5 min at 660× *g*, resuspended in 50 μL buffer B (250 mM Tris HCl pH 7.8, 60 mM KCl, 1 mM DTT, 2 mM PMSF, and 20 % *v*/*v* glycerol in PBS) and centrifuged at 4 °C for 15 min at 9500× *g*. The cell pellets were resuspended in 100 μL buffer C (10 mM Hepes pH 7.9, 1 mM EDTA pH 8.0, 60 mM KCl, 1 mM DTT, 1 mM PMSF, and protease inhibitors) and centrifuged at 4 °C for 5 min at 660× *g*. The samples were then washed twice with 1 mL buffer C, resuspended in 50 μL buffer B, and exposed to 3 cycles of freeze/thawing. Finally, the samples were centrifuged at 4 °C for 15 min at 9500× *g*, and nuclear proteins were obtained.

### 3.11. RNA Isolation and Quantitative Real-Time-PCR (qRT-PCR)

Total RNA was isolated. Total RNA was isolated from cultured using Trizol Reagent (Life Technologies, Grand Island, NY, USA) according to the manufacturer instructions and spectrophotometrically quantified cells and assessed by agarose gel electrophoresis. Then, cDNA was generated using 3 μg of RNA and real-time PCR was performed with Light-Cycler^®^ 480 (Roche Diagnostics GmbH, Mannheim, Germany) using SYBR Green I Master Mix (Life Technologies). The forward and reverse primers were used at the concentration of 10 mM. Reactions were carried out starting with a 10 min step at 95 °C; subsequently 40 cycles—consisting of a 10 sec of denaturation at 95 °C, 5 s of annealing at 56 °C and 15 s of extension at 72 °C—were performed. As an internal standard was selected the 18S RNA. To quantify specific mRNAs, real-time PCR assays were conducted using the following primer sets: forward Akt: 5′-TCT ACA CCC ACA GAT GAC AG-3′; reverse Akt: 5′-CTC AAA TGC ACC CGA GAA AT-3′; forward Bcl-2: 5′-GGA AGT GAA CAT TTC GGT GAC-3′; reverse Bcl-2: 5′-CTC CAT CAG CTT CCA GAC AT-3′; forward 18S 5′-CGA TGC TCT TAG CTG AGT GT-3′; reverse 18S 5′-GGT CCA AGA ATT TCA CCT CT-3′.

### 3.12. Cell Cycle and Apoptosis Analysis

HeLa cells were seeded at 5.0 × 10^5^ in 100 mm culture dishes. The day after, the cells were treated with DMSO and **12** at 10 μM and 20 μM for 48 h. The cell cycle was evaluated by propidium iodide (PI) staining of permeabilized cells (BD FACSCalibur flow cytometer, Becton Dickinson, San Jose, CA, USA). Data from 5000 events per sample were collected. The percentages of the elements in the hypodiploid region and in G0/G1, S, and G2/M phases of the cell cycle were calculated using the CellQuest and MODFIT software, respectively. The apoptosis analysis was carried out according to the protocol described in Annexin V, FITC Apoptosis Detection Kit (Dojindo EU GmbH, Munich, Germany). The samples were detected by Flow Cytometry (BD FACSCalibur flow cytometer, Becton Dickinson, San Jose, CA, USA) and analyzed by BD FACSuite software.

### 3.13. Statistical Analysis

All the experiments were conducted at least in triplicate and the measurements were carried out twice (*n* ≥ 6). Obtained data were reported as the mean values ± SD. Student’s t-tests were performed to estimate the statistical significance of the measured differences between treatment groups. They were considered significant when *p* < 0.05. 

## 4. Conclusions

One of the most important challenges of modern biochemistry is to provide validated information for orientating drug design and development. This involves defining the mechanism of action of effective bioactive compounds and identifying compounds that can interact with “druggable” proteins, modulating their activity and leading to a therapeutic effect. Once an effective interactor of a target protein has been found, a characterization of the binding mode and the structural and functional modification induced on the protein can be achieved. 

Here, we described the ability of trachylobane diterpene 6,19-dihydroxy-ent-trachiloban-17-oic acid (**12**) to interact with hNcl inside cancer cells and inhibit its mRNA chaperone activity and prevent phosphorylation. Based on the experimental and computational data, a detailed description of the binding of that compound to hNcl was proposed, and the resulting model is in good agreement with the observed biological. 

Interestingly, from our study emerged that, although compound **12** was selected because it shares some structural features with oridonin, the effects of the interaction of these two compounds with hNcl were clearly different. Indeed, while oridonin inhibited both the mRNA-chaperone action of hNcl and the key role that this protein plays in protein synthesis, **12** did not affect the latter activity. Conversely, only **12** markedly reduced post-transcriptional modifications of hNcl. These observations indicated **12** to interfere mainly with hNcl in pathological conditions, where the actions of phosphorylation and mRNA-chaperone are more relevant, thus suggesting an effect of this compound mainly towards tumor cells. In agreement with this result, **12** exerts a significant pro-apoptotic action against HeLa cells, while no toxicity against healthy cells was observed. Taken together, the reported results indicated **12** as a putative anticancer agent or as a promising lead compound for setting up and optimizing new hNcl inhibitors. Moreover, the results obtained showed once again how the molecules of natural origin are an irreplaceable source of biological activities; however, for the great potential of these molecules to be effectively usable, their mechanism of action must be understood and described at the molecular level.

## Figures and Tables

**Figure 1 ijms-23-11390-f001:**
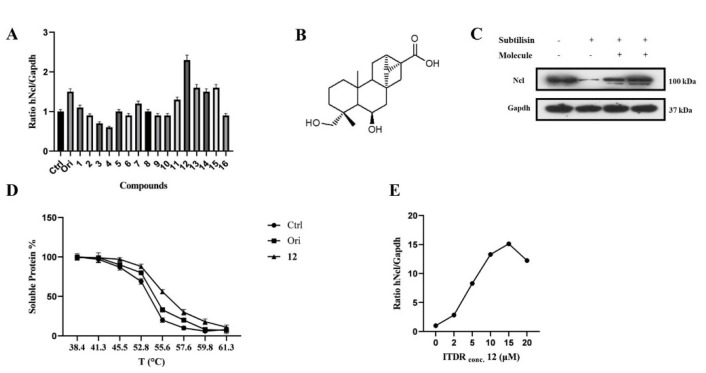
6,19−dihydroxy−ent−trachiloban−17−oic acid (compound **12**) interacts effectively with hNcl. (**A**) Results of the CETSA−based screening of a small library of diterpenes. (**B**) Chemical structure of compound **12**. (**C**) DARTS experiment results. (**D**) Thermal denaturation curve of hNcl in HeLa cell lines untreated or incubated with oridonin or compound **12**. (**E**) CETSA-based measurement of the EC_50_ of compound **12** in HeLa cells.

**Figure 2 ijms-23-11390-f002:**
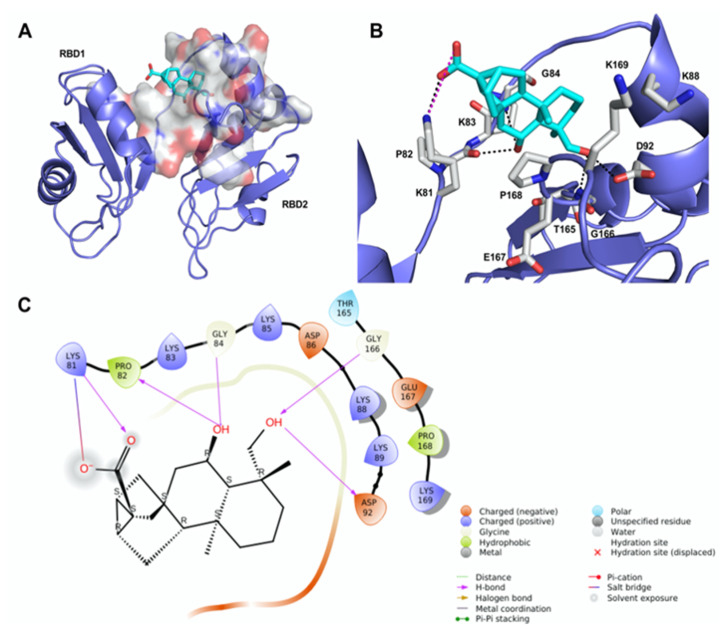
Theoretical binding mode of compound **12** to RNA-binding domains 1 and 2 (RBD1,2) of nucleolin. (**A**) The best-scored pose of compound **12** (cyan sticks) within RBD1,2 represented as a slate ribbon model. The site identified by SiteMap is represented as a white Connolly surface. (**B**) Close-up view of compound **12** docked pose. Only amino acids located within 4 Å of the bound ligand are displayed (white sticks) and labeled. The H-bonds discussed in the text are depicted as dashed black lines; the salt bridge is depicted as a dashed magenta line. (**C**) Two-dimensional ligand interaction diagram of compound **12**’s docked pose.

**Figure 3 ijms-23-11390-f003:**
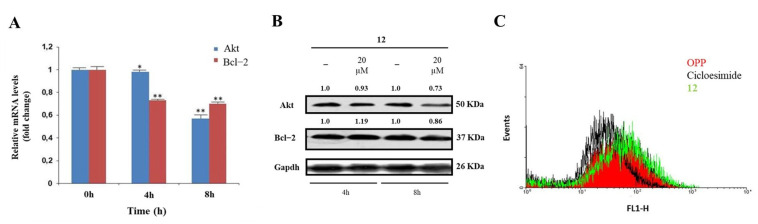
Effects of interaction with compound **12** on hNcl bioactivity. (**A**) Results of real−time PCR of the Akt and Bcl−2 mRNAs following HeLa cell treatment with compound **12**. * *p* < 0.05 and ** *p* < 0.01 vs. time 0 h. (**B**) Western blot analysis of Akt and Bcl−2. (**C**) Protein neosynthesis in HeLa cells incubated with compound **12**. The numbers shown above the blots are representative of the densitometric analysis after normalization.

**Figure 4 ijms-23-11390-f004:**
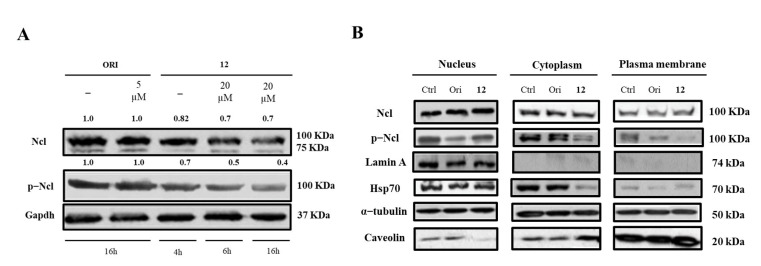
Nucleolin (Ncl) and phosphorylated Ncl (p−Ncl) levels. Western blot analysis of Ncl and p−Ncl in whole Hela cells (**A**) and in different cell compartments (**B**). The numbers shown above the blots indicate the results of densitometric analysis after normalization.

**Figure 5 ijms-23-11390-f005:**
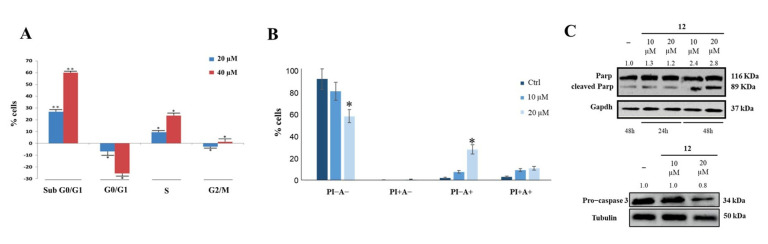
Effect of compound **12** on cell cycle and apoptosis. (**A**) HeLa cells incubation with compound **12** induced concentration-dependent arrest of the cell cycle in subG0/G1. (**B**) Compound **12** induced early apoptosis of HeLa cells. * *p* < 0.05 and ** *p* < 0.01 vs. Ctr. (**C**) Parp and pro−Caspase 3 cleavage observed following HeLa cells incubation with compound **12**. The numbers shown above the blots are representative of the densitometric analysis after normalization.

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
