# Peer review of "Impairment of Nucleolin Activity and Phosphorylation by a Trachylobane Diterpene from *Psiadia punctulata* in Cancer Cells"

_ijms, 2022, doi:10.3390/ijms231911390_

Round 1
Reviewer 1 Report
The study by Maria Laura Bellone et al. provided new insights into a potential therapy by targeting human nucleolin (hNcl). Authors reported a compound called 6,19-dihydroxy-ent-trachiloban-17-oic acid (compound 12) as a potential inhibitor of human nucleolin (hNcl) protein. They also declared compound 12 can induce apoptosis and cell cycle arrest of Hela cell line. Although the paper may be of some interest, various critical issues are present in the submitted manuscript that need to be overcome.
Major comments:
1. Compound 12 showed the best cytotoxicity to Hela cells in the MTT assay, and Hela cells were also used in subsequent experiments. If authors want to compare the cytotoxicity between normal and cancer cells, they should choose another normal cell line. Because Hela is adherent cell line and Jurket, as a suspension cell line, may not suitable.
2. In figure 3A, if authors meant to show the apoptosis-related gene change, they should examine the change of p-Akt and perform series of assay to determine the Bcl-2-related mitochondria apoptosis such as JC-1 staining, caspase assay or the protein change of Cytochrome C.
3. In figure 4A, the protein expression of Ncl at 100kDa had significantly decreased by adding compound 12. If the compound 12 directly bound to Ncl, the expression of Ncl should not change. Please explain. Moreover, combined with the binding model in figure 2, authors should explain why compound 12 bind to the RNA binding domain but inhibited the phosphorylation of Ncl.
4. Also in figure 4A, authors should explain the meaning of “1.0, 1.0, 0.82…..”in the figure legends. Whether the numbers stand for percentage or concentration of compounds, authors should explain why they compared Ncl and p-Ncl under different condition of treatment.
5. The cell cycle arrest assay in figure 5A requires flow cytometric results, not just statistical data. In figure 5B, a statistical graph of the flow apoptosis results should be shown to increase the credibility of the experimental data.
6. Authors should add more analysis methods to prove their theory, such as immunofluorescence assay, point mutation or ELISA assay. The workload of this manuscript is too little to support their conclusion.
Minor comments:
1. Two "AS1411" words appear in line 131 of the article. Please check.
2. Authors did not describe the MTT assay in Material and Method part. And the time of compound 12 treatment for cytotoxicity was not mentioned.
3. The uncropped WB blots of Figure5B were not provided. The tendency of GAPDH was confusing.
4. In line 191, authors declare the treating time were 4, 6 and 16h. However, in figure 4, they labeled 4, 8 and 16h. Meanwhile, in part 2.4 of the article, the time of action of compound 12 marked in figure 3A is 3h and 6h, while the time stated in the article is 4h and 8h, which should be checked.
5. Authors should use a more scientific graph processing tools like GraphPad or Origin instead of Excel, especially in figure 1.
Reviewer 2 Report
The manuscript with its rather elaborate title "Impairment of nucleolin activity and phosphorylation by trachylobane diterpene 6,19-dihydroxy-ent-trachiloban-17-oic acid from Psiadia punctulata induces apoptosis of cancer cells" by Bellone, M.L., Fiengo, L., De Tommasi, N., Piaz, F.D. et al. delineates the biological activity of the title diterpene isolated from a herb used in traditional medicine against human nucleolin, and its ability to affect phosphorylation pathway. As in a previous case involving the structurally disparate kaurane diterpene oridonin (ref. 24), the compound designated compound 12 in the manuscript is shown to inhibit human nucleolin (hNcl), which plays vital roles, inter alia, in protein construction pathways including ribosomal synthesis. The authors then go on to demonstrate attenuation of mRNA activity coupled with reduced capacity to initiate phosphorylation. Studies conducted in vitro on HeLa cells wherein apoptotic cell death associated with cell cycle arrest at the G0/G1 phase is observed. No direct association of apoptosis with hNc1 inhibition was demonstrated, although this was assumed to be the underlying cause for cell cycle arrest.
The MS is well written, and relatively easy to follow, and given the import of the work, is recommended for publication. However, there are some issues. The MS complements the authors' elegant earlier work on the structurally rather different diterpene oridonin (ref. 24) (used as a positive control in the current work) wherein it is demonstrated that this compound also interacts with nucleolin. However, the authors are aware of alternative pathways for antitumour mechanism of action for this compound class (cf. also Sun, Lou et al. ent-Kaurane diterpenoids induce apoptosis and ferroptosis through targeting redox resetting to overcome cisplatin resistance; Redox Biology 43 (2021) 101977; doi: 10.1016/j.redox.2021.101977). One may note that as oridonin has a chemically reactive a,b-unsaturated ketone associated with the D-ring, it is indeed structurally and chemically distinct to the trachylobane 12 in the current MS. It is not structurally homologous, and it is not correct to assert this in the MS. Thus, as both compounds interact with nucleolin, the question must arise as to whether such an interaction is responsible for antitumour activity. To be sure, interference with m-RNA chaperone activity is likely to be important, but that this may lead to apoptotic cell death was not demonstrated in the current MS.
The foregoing also bears on the title of the MS. This is unfortunately rather cumbersome and in incorporating a quasi-systematic name for the title compound doesn't add to the impact. Why not something like "Inhibition of nucleolin activity and phosphorylation in tumour cells by a trachylobane diterpene from Psiadia punctulata "? As the authors do not explicitly demonstrate that such inhibition leads to apoptotic cell death, the rest of the title may not be appropriate. Whilst the modelling precepts in Figs 2A and 2B appear to be all right, the Fig 2C is of poor quality, and needs to be redrawn to show clearly the salt bridge between the primary terminal amino group of lysine and the carboxyl group and the H-bonding interactions involving the other amino acids and the H-bond donor and acceptor groups in compound 12. Such a docking pose also raises the question of how the structurally distinct oridonin would interact non-covalently with nucleolin; the studies discussed in ref 24 were remarkably thorough and of high quality, but no modelling experiments were discussed. Overall, the question arises as to whether promiscuous binding of these structurally relatively complex compounds to nucleolin occurs, and whether in fact such binding is explicitly associated with antitumour activity. Clearly more work is required for future publications to assess whether indeed the compounds may serve the basis for development of selective hNcl inhibitors.
On this basis then, the MS must be published, but caveats indicating the foregoing concerns be added. Other minor grammatical aspects must be attended to, and the figures improved to render them legible.
Round 2
Reviewer 1 Report
Authors'reply is appropriate.